# Treatment Strategies for KRAS-Mutated Non-Small-Cell Lung Cancer

**DOI:** 10.3390/cancers15061635

**Published:** 2023-03-07

**Authors:** Éabha O’Sullivan, Anna Keogh, Brian Henderson, Stephen P. Finn, Steven G. Gray, Kathy Gately

**Affiliations:** 1Thoracic Oncology Research Group, Department of Clinical Medicine, Trinity Translational Medicine Institute, St. James’s Hospital, D08 W9RT Dublin, Ireland; 2Thoracic Oncology Research Group, Laboratory Medicine and Molecular Pathology, Central Pathology Laboratory, St. James’s Hospital, D08 RX0X Dublin, Ireland

**Keywords:** KRAS, G12C mutation, NSCLC, IMAs, sotorasib, adagrasib, drug resistance, BRAF, ACOX2

## Abstract

**Simple Summary:**

KRAS plays an important role in transmitting signals from growth factors on the outside of the cell to the cell nucleus. It regulates cell proliferation, growth, and survival. The activation of KRAS occurs in multiple tumour types, either directly due to a mutation in the KRAS gene or indirectly via other proteins in the pathway. KRAS was considered an undruggable protein due to its smooth surface, but a recent discovery of a specific pocket in its structure has led to the development of several inhibitors that target the G12C mutation. Two of these, sotorasib and adagrasib, have been approved in advanced non-small-cell lung cancer, and others are currently being tested in clinical trials. Cancer cells can limit the effect of KRAS G12C inhibitors by switching on other proteins or through the development of new resistance mutations; therefore, these inhibitors will likely be used in combination with other therapies to treat patients more effectively.

**Abstract:**

Activating mutations in KRAS are highly prevalent in solid tumours and are frequently found in 35% of lung, 45% of colorectal, and up to 90% of pancreatic cancers. Mutated KRAS is a prognostic factor for disease-free survival (DFS) and overall survival (OS) in NSCLC and is associated with a more aggressive clinical phenotype, highlighting the need for KRAS-targeted therapy. Once considered undruggable due to its smooth shallow surface, a breakthrough showed that the activated G12C-mutated KRAS isozyme can be directly inhibited via a newly identified switch II pocket. This discovery led to the development of a new class of selective small-molecule inhibitors against the KRAS G12C isoform. Sotorasib and adagrasib are approved in locally advanced or metastatic NSCLC patients who have received at least one prior systemic therapy. Currently, there are at least twelve KRAS G12C inhibitors being tested in clinical trials, either as a single agent or in combination. In this study, KRAS mutation prevalence, subtypes, rates of occurrence in treatment-resistant invasive mucinous adenocarcinomas (IMAs), and novel drug delivery options are reviewed. Additionally, the current status of KRAS inhibitors, multiple resistance mechanisms that limit efficacy, and their use in combination treatment strategies and novel multitargeted approaches in NSCLC are discussed.

## 1. Introduction

Kirsten rat sarcoma viral oncogene homologue (KRAS) is the best-known oncogene with the highest mutation rate among all cancers. KRAS was first detected in 1982 in lung cancer cells, located on the short arm of chromosome 12 (12p11.1–12p12.1) [1]. It is a member of the RAS family of GTPase signal transducer proteins, which hydrolyse guanosine triphosphate (GTP) to guanosine diphosphate (GDP) and includes the Harvey rat sarcoma viral oncogene (HRAS) and neuroblastoma rat sarcoma viral oncogene (NRAS). RAS proteins are molecular switches that, under normal physiological conditions, cycle between the inactive GDP-bound state and the active GTP-bound state to transduce extracellular signals to the interior of the cell (Figure 1). RAS proteins interact to form functional clusters on membranes and efficiently recruit downstream effectors [2]. Structurally, KRAS is divided into an effector-binding lobe, an allosteric lobe, and a carboxy-terminal region responsible for membrane anchoring. The effector lobe comprises the P loop and the switch I/switch II loop regions. The cycling between the inactive and active form causes a conformational change in the switch I and II regions [3], which plays a critical role in KRAS downstream signalling through mediating protein–protein interactions with effector proteins that include RAF1 in the MAPK pathway or PI3K in the PI3K–AKT pathway. Thus, activated KRAS regulates several cellular processes such as differentiation, proliferation, and apoptosis [4].

KRAS mutations are an early event in lung tumorigenesis, associated with a history of smoking [5,6], a high mutation burden, and elevated markers of immune evasion (PD-L1 and PD-L2) [7]. KRAS-driven tumours shift their metabolism towards a more anabolic profile by upregulating the expression of multiple rate-limiting enzymes involved in key metabolic processes essential for survival. This metabolic reprogramming promotes glycolysis and lactate production [8,9] through increased glucose transporter 1 (GLUT1) expression and rate-limiting glycolytic enzymes, including hexokinases, phosphofructokinase 1 (PFK1), and lactate dehydrogenase A (LDHA) [9]. KRAS-driven tumours produce key lipid mediators establishing an immunosuppressive tumour microenvironment (TME) and utilise exogenous lipids produced by the TME such as fatty acids (FAs), prostaglandins, and other lipid mediators that sustain tumour growth and metastasis.

Mutated KRAS was notoriously challenging to target and even “undruggable” due to its smooth, spheric structural biology, and a lack of drug-binding pockets, which limited therapeutic interventions. Four decades of research finally culminated in the first major breakthrough in the race to target KRAS-driven cancers. In 2013, a seminal breakthrough by the Shokat lab showed that the activated KRAS isozyme, caused by the G12C mutation in the KRAS gene, can be directly inhibited via a newly identified switch II pocket [10]. This discovery has led to the development of a new class of selective small-molecule inhibitors against the KRAS G12C isoform. In vitro, preclinical and clinical trial data demonstrated antitumor activity and clinical efficacy. Sotorasib (Lumakras, Amgen) monotherapy was approved in 2021 as a second-line treatment in KRAS G12C-driven, locally advanced, or metastatic non-small-cell lung cancer (NSCLC). Adagrasib (Mirati Therapeutics) was recently FDA-approved (December 2022) for accelerated approval as a treatment for patients with previously treated KRAS G12C-positive NSCLC. Such is the interest in this target that at least twelve others are currently under investigation in clinical trials (Table 1). While single-agent efficacy has been shown, reports have highlighted that acquired drug resistance emerges quickly, with multiple mechanisms emerging together. Sequential and combination treatment approaches are more likely to lead to superior and more durable outcomes. The development of directly targeted agents to the myriad of other KRAS mutations remains a challenge. However, the US FDA’s recent IND clearance (January 2023) of MRTX-1133 (Mirati Therapeutics), a small molecule that selectively targets the KRAS G12D allele, is encouraging, and trial results are eagerly awaited. This review examines KRAS mutation prevalence, subtypes, and rates in treatment-resistant invasive mucinous adenocarcinomas (IMAs) and novel drug delivery options. The current status of KRAS G12C-specific inhibitors, the emerging acquired resistance mechanisms, new pan-KRAS therapeutics, and novel combination strategies are also discussed.

## 2. Oncogenic KRAS Mutations

### 2.1. KRAS Mutations: Types and Prevalence

KRAS mutations are present in approximately 25% of all cancers, making them the most common oncogenic driver [11]. Approximately 5% of small-cell lung cancer (SCLC) [12] and 39% of NSCLC (Figure 2a) tumours harbour KRAS mutations [13]. Other than NSCLC, pancreatic adenocarcinomas (PDACs) and colorectal carcinomas (CRCs) commonly harbour KRAS mutations with prevalence rates of >90% and 40%, respectively. The dominant missense point mutation differs across cancers; for example, KRAS G12D is the dominant mutant subtype in pancreatic adenocarcinoma at 67.6%, and G12D and G12V are the most common mutations in colorectal carcinoma, representing 39% and 24%, respectively [14]. The rates of G12C mutations vary vastly across cancers with rates in NSCLC, CRC, and PDAC of 39%, 7%, and 1%, respectively [11]. In NSCLC, G12C mutations are higher in females (43.4%) than in males (32.7%) (Figure 2b,c) [13].

Currently, treatments are focused on G12C mutations, and therefore, there is little if any role for these targeted therapies in PDAC. The initial clinical trials investigating G12C inhibitors in CRC (AMG510 and MRTX849) showed promise; however, these clinical studies unexpectedly reported limited response rates [15]. The reason for this possible resistance is thought to lie within the intrinsic differences between NSCLC cells and CRC cells, even when harbouring the same driver mutation. Unlike in NSCLC, CRC cell lines have high basal receptor tyrosine kinase activation and are responsive to growth factor stimulation [16]. G12C inhibition has been shown to induce higher phospho-ERK rebound in CRC cells than in NSCLC cells, which may underlie this resistance [16]. However, combining EGFR- and G12C-targeting treatments has been shown to be highly effective in preclinical studies [16] and may be utilised as a mechanism for combating the resistance that inevitably occurs in NSCLC.

Other less common cancers that have high rates of KRAS mutations include appendiceal mucinous adenocarcinomas with rates of 60–80% and ovarian mucinous carcinomas with rates of 64% [17]. Cancers that harbour KRAS mutations share many common features. KRAS mutations are associated with smoking, with an incidence of 25–35% in smokers and 5% in non-smokers [11]. However, this tends to be mutation-dependent, with G12D mutations more often found in non-smoking patients (Figure 2d) and G12C mutations found in former and current smokers (Figure 2e,f) [13,18].

### 2.2. Genetic Alterations Co-Occurring with KRAS Mutations

The co-occurring alterations of other genes add to the heterogeneity of KRAS-mutated tumours and influence biological behaviours [19], clinical outcomes, and response to treatment [20,21]. Three major subsets include STK11/LKB1 (KL), TP53 (KP), and CDKN2A/B inactivation (KC) [19]. Another critical co-mutation is KEAP1, which is included in the KL subgroup. STK11/LKB1 tumours are associated with a lower expression of immune markers (e.g., PD-L1) if KEAP1 is co-mutated, correlating to resistance to PD-1 blockade in KRAS-mutated lung adenocarcinoma [22]. A study in advanced KRAS-mutated lung cancer by Arbour et al. showed that KEAP1 co-mutations are independent prognostic markers for poorer survival (HR = 1.96; *p* < 0.001) and are associated with less response to chemotherapy (HR = 1.64; *p* = 0.03) and immunotherapy (HR = 3.54; *p* = 0.003) [23]. An increase in somatic mutations, inflammatory and immune checkpoint markers, and prolonged relapse-free survival were observed in TP53 tumours. The two-hit inactivation of CDKN2A and CDKN2B frequently occurs in mutant KRAS lung adenocarcinoma and results in suppressed mTORC1 signalling [24]. These data highlight the importance of the molecular profiling of all tumours to enable the stratification of patients to appropriate targeted treatments.

## 3. KRAS a Therapeutic Target in Treatment Resistant Lung Adenocarcinomas (LUAD)

### 3.1. Histological Patterns Associated with Driver Mutations in Lung Cancer

Lung cancers, particularly lung adenocarcinomas (LUADs), are heterogenous and include various histological subtypes and molecular alterations that impact chemosensitivity as well as overall survival [25]. LUADs consist of non-mucinous adenocarcinomas and invasive mucinous adenocarcinomas (IMAs) (Figure 3), representing approximately 90% and 10% of cases, respectively [26]. Invasive non-mucinous carcinomas are further subdivided into five histological patterns: lepidic, acinar, papillary, micropapillary, and solid [27]. These are strongly associated with prognosis, with lepidic having the most favourable, acinar and papillary having intermediate, and solid and micropapillary patterns having the worst prognosis [28].

The prognosis of IMAs in the lung on the other hand is less well characterised, with several studies demonstrating conflicting results [26,28]. Associations between histology and driver mutations have also been made. EGFR mutations are associated with lepidic predominant LUADs [29], ALK and ROS rearrangements with cribriform pattern and signet-ring features [30,31], and KRAS mutations showing a strong association with IMAs [32].

### 3.2. Associations between KRAS Mutations and Invasive Mucinous Lung Adenocarcinomas

The invasive mucinous adenocarcinomas (IMAs) of the lung are rare, representing 3–10% of LUAD cases [26]. IMAs have goblet cell morphology, with abundant intracytoplasmic mucin [27]. These tumours develop through distinct genetic pathways that differ from those of non-mucinous LUADs. KRAS mutations have a strong association with IMAs, with a prevalence of 60% [32]. Conversely, features seen in non-mucinous LUADs such as TP53 mutations, high tumour mutational burden (TMB), and targetable mutations such as EGFR, ALK, and BRAF v600E are rarely seen [33]. In addition to having distinct genetic features, IMAs differ in response to conventional treatments, even though they are treated similarly to non-mucinous LUADs. For example, patients receiving non-TKI platinum-based chemotherapy with stage IV IMAs have similar overall survival (OS) to untreated IMA patients [34]. Reduced repose rates to chemotherapy are also seen in other cancers with mucinous histology such as mucinous colorectal carcinomas and mucinous ovarian carcinomas [35].

The reduced sensitivity of IMAs to chemotherapy has not been fully elucidated but may be due to the viscous nature of mucus, which may hinder drug delivery, reducing therapeutic efficacy [36]. Additionally, mucinous carcinomas may have poorer vasculature, also leading to the decreased delivery of drugs [37]. Mucins are heavily glycosylated proteins, responsible for the gel-like, cohesive, and adhesive nature of mucus. There are 21 mucin-related genes whose expression and functions vary between tissues. Within lung cancer, MUC2 and MUC5A are the most common [38]. 

To our knowledge, the exact mechanism of how a gain-of-function mutation in KRAS relates to mucin production in these tumours is unclear; however, it may lie in the fact that MUC2 and MUC5AC are target genes for EGFR ligands in lung cancer cells, and the upregulation of these two genes leads to the activation of the EGFR/Ras/Raf/MAPK kinase signalling pathway [39,40]. How this histological subtype, with abundant mucin, may affect the delivery of novel KRAS inhibitors, to our knowledge, has not been answered; however, this may lead to a number of patients having lower rates of response than expected. Developing targeted therapies to penetrate mucin for adequate drug delivery, as well as developing strategies to overcome resistance pathways, are all needed to combat the cancers harbouring these mutations.

### 3.3. Novel Therapeutic Delivery Approaches to Target KRAS-Mutated NSCLC

Patients with IMAs have been shown to have lower rates of partial response to chemotherapy than those with non-mucinous LUADs [34]. One potential explanation for these lower response rates is poor drug delivery through the abundant viscous mucin present in this subtype of LUADs. The development of drugs that can penetrate this mucin barrier, therefore, may be a fruitful therapeutic approach.

Advances in nanomedicine have generated a broad range of engineered nanoparticles (NPs) for drug delivery applications, which are designed to promote drug transport across cell membranes and to deliver drugs in a controlled and targeted manner. The ability of a drug to penetrate mucin relies on its small size and a neutral highly hydrophilic surface [41]. NPs are synthetic particles of <100nm, and therefore they represent an attractive method to penetrate mucin [42]. NP Liposomes are composed of an outer phospholipid bilayer surrounding an aqueous core, which allows for the transport of both hydrophilic and lipophilic drugs, as well as drugs with a wide range of size and charge [43]. They are biodegradable and biocompatible and present low immunogenicity; however, they tend to degrade rapidly and adhere together [44]. Polyethylene glycol (PEG), a hydrophilic and non-ionic polymer, when coated on the surface of NPs, confers more stability and allows rapid diffusion through mucin by reducing particle adhesions to the mucin fibres found in the mucus mesh [45]. PEG has become popular with research groups engineering mucous-penetrating drug carriers [45]. CALYPSO, a Phase III study, compared carboplatin–PEGylated liposomal doxorubicin with carboplatin–paclitaxel in patients with ovarian cancer. Treatment led to delayed progression and similar overall survival compared with the carboplatin–paclitaxel group [46]. To our knowledge, no clinical trials have studied PEGylated liposomes to specifically treat mucinous carcinomas; however, they may represent an attractive target to penetrate mucin in these cancers.

Targeted NP-containing docetaxel (BIND-014) has also been studied as a second-line therapy for patients with KRAS-positive or squamous cell NSCLC [47]. BIND-014 is approximately 100 nm in size and is composed of docetaxel encapsulated in a polymer core made of hydrophobic poly(lactide) surrounded by a hydrophilic poly(ethylene glycol) conjugated with a small molecule of PSMA-targeting ligands [47]. In a Phase I study, (NCT02283320) BIND-014 was well tolerated, with predictable and manageable toxicity and tumour shrinkage at doses below the conventional docetaxel formulation dose [47].

Additionally, liposomes can be used to deliver therapeutic agents to inhibit specific oncogenes. Ozeplasmid (Reqorsa), a non-viral lipid nanoparticle, encapsulates a plasmid with TUSC2, a tumour-suppressor gene, and in combination with osimertinib is currently in Phase I and II clinical trials (NCT04486833) in advanced lung cancer patients who progressed on osimertinib (Acclaim-1). Ozeplasmid has recently been approved by a safety review committee (SRC), according to Genprex, Inc. (Austin, TX, USA) [48], confirming its favourable safety profile. The final enrolment for this clinical trial is expected to be completed in the first quarter of 2023. Other NPs currently being investigated in LUADs include magnetic, polymer, liposomes, solid lipid, metal, and viral NPs and are summarised elsewhere [49].

## 4. Therapeutic Options for KRAS Mutated NSCLC 

### 4.1. Targeting KRAS

KRAS can be targeted indirectly by the inhibition of upstream regulators such as SOS or downstream targets such as MEK or PI3K. A Phase I clinical trial NCT04111458 is currently examining the efficacy of SOS inhibitor (BI1701963) monotherapy and its combination with MEK inhibitor trametinib.

The direct targeting of KRAS has proven challenging due to its picomolar affinity for GTP, the lack of suitable pockets for high-affinity small-molecule binding, and its high mutation rate [50]. However, in recent years, our understanding of KRAS has significantly increased with a resurgence of publicly available KRAS structures and increased computational capacity, enabling molecular dynamic (MD) simulations to study the dynamics of KRAS protein in more detail at the atomistic level [51].

The breakthrough discovery by the Shokat laboratory identified a new allosteric binding site on KRAS known as its ‘switch II pocket’(S-IIP) [10]. This demonstrated the possibility to inhibit inactive KRAS G12C with covalent binders, identified by screening a library of roughly 500 fragment-like disulphides [10]. This led to the development of twelve KRAS G12C-directed inhibitors, all tested in clinical trials and two approved for use in the clinic (Table 1).

### 4.2. Direct Inhibitors of KRAS G12C

#### 4.2.1. Sotorasib (AMG510)

Amgen’s KRAS G12C covalent inhibitor research program identified and developed the clinical candidate drug sotorasib, which entered clinical trial in August 2018 CodeBreaK100 (Phase I) (NCT03600883) [52,53]. Sotorasib binds to the cysteine of the switch II region (S-IIP), keeping KRAS in its inactive GDP-bound form, inhibiting KRAS signalling, and suppressing the MAPK pathway. This compound was the first to show the benefit of KRAS G12C inhibitor treatment on disease progression in patients, demonstrating a 37.1% response rate, a progression-free survival of 6.8 months, and a median overall survival of 12.5 months in a Phase II clinical trial of 126 patients with advanced NSCLC CodeBreaK100 (Phase II) [53]. Sotorasib was the first KRAS inhibitor to receive FDA approval on 28 May 2021 [54,55]. Qiagen therascreen KRAS RGQ PCR kit and Guardant360 CDx were also approved for use to screen for the G12C mutation in a person’s tumour tissue or blood, respectively. Ongoing clinical trials are investigating sotorasib as monotherapy or in combination with various anticancer agents in patients with advanced or metastatic solid tumours.

#### 4.2.2. Adagrasib (MRTX849)

Adagrasib (MRTX849), developed by Mirati Therapeutics, is another direct small molecule that targets KRAS G12C and entered the KRYSTAL-1 clinical trial in January 2019 (NCT03785249) [56]. The FDA recently approved adagrasib for accelerated approval. Adagrasib also binds to the cysteine of the S-IIP-inhibiting KRAS-dependent signalling including the suppression of the MAPK pathway [57]. In a panel of human KRAS G12C cell line (CDX) and patient-derived xenograft (PDX) models, Hallin et al. demonstrated tumour regression exceeding 30% volume reduction from baseline in 17 out of 26 models (65%) at approximately three weeks of treatment [58]. Riely et al. demonstrated that patients with STK-11 co-mutations who typically have a relatively poor response to immune checkpoint inhibitors had an ORR of 64%. Interestingly, these patients had a minimal expression of CD4 and CD8 transcripts at baseline, and after treatment with adagrasib, these transcripts increased, suggesting a potential immune response to therapy [59]. Ongoing clinical trials are investigating adagrasib as monotherapy or in combination with other anticancer agents in patients with advanced or metastatic solid tumours.

#### 4.2.3. Other Direct KRAS G12C Inhibitors

LY3499446 (Eli Lily) entered a clinical trial in November 2019 (NCT04165031) but was terminated due to unexpected toxicity. Toxicity can still be an issue despite the mutant-specific nature of inhibitors and may be due to off-target effects on other cysteine-containing proteins.

Newer inhibitors in early-phase testing include GDC-6036 (Genentech), which is being investigated in a Phase I clinical trial as monotherapy and in combination with other anticancer therapies in patients with advanced/metastatic solid tumours with a KRAS G12C mutation (NCT04449874). D-1553, developed by InvestisBio, entered clinical trials in October 2020 as monotherapy (NCT04585035). JDQ443 (Novartis Pharmaceuticals) is being examined in a Phase I/II clinical trial as monotherapy and in combination with TNO155 (SHP2 inhibitor) and/or spartalizumab (anti-PD1 antibody) in patients with advanced or metastatic solid tumours with the KRAS G12C mutation (NCT04699188). BI 1,823,911 (Boehringer Ingelheim) is a small-molecule KRAS G12C inhibitor that entered clinical trials in July 2021 (NCT04973163). LY3537982 (Eli Lily) entered a Phase I/II clinical trial (NCT04956640) as monotherapy or in combination with other targeted therapies, as outlined in Table 1. JAB-21822 (Jacobio) is currently being investigated in several clinical trials as monotherapy or in combination with Cetuximab (EGFR mAb) or JAB-3312 (SHP2 inhibitor) (Table 1). YL-15293 (Shanghai YinLi) is being examined as monotherapy in a clinical trial (NCT05119933). Finally, RMC-6291 (Revolution Medicine) is currently under investigation as monotherapy in a clinical trial (NCT05462717).

#### 4.2.4. Beyond KRAS G12C Inhibitors

Mirati Therapeutics developed a small-molecule MRTX-1133 that selectively targets the KRAS G12D allele [60] and has received IND clearance by the US FDA (Jan 2023), enabling Phase I initiation for a first-in-class oral inhibitor. KRAS G12D inhibitors will benefit never smokers with NSCLC for whom the mutation rate is 55.7% (Figure 2d) and could be a game-changer in PDAC and colorectal carcinoma, where the mutation rates are 67.6% and 39%, respectively.

Proteolysis-targeting chimeras (PROTACs) are bifunctional molecules that bind to both the target of interest and an E3-ligase protein, connected by a linker, degrading the protein of interest via the cellular proteasomal degradation machinery [61]. PROTACs have emerged as a new class of targeted therapy, and several groups have developed specific KRAS G12C PROTACs incorporating specific inhibitors such as ARS1620 [62] or MRTX849 [63]. Boehringer Ingelheim reported on a direct pan-KRAS inhibitor and direct pan-KRAS PROTAC that inhibits all the major KRAS mutants while sparing HRAS and NRAS [64].

Other targeted strategies include a combination of SOS inhibitor BI1701963 and MEK inhibitor trametinib (NCT04111458) [65]. This ensures both the upstream inhibition of the binding of SOS (GEF) to KRAS and the downstream inhibition of MEK, which effectively shuts downs the signalling of the MAPK pathway.

## 5. Resisting KRAS Targeted Therapy

### 5.1. Resistance to KRAS G12C-Targeted Therapy

KRAS G12C inhibitors have demonstrated impressive activity and response rates. However, similar to other targeted therapies, these inhibitors are also plagued by both intrinsic and acquired resistance mechanisms, which limits their efficacy and duration of response in patients. These mechanisms are outlined below and are all-encompassing often with several emerging in parallel within a tumour.

#### 5.1.1. Intrinsic Resistance

KRAS activation stimulates signalling through both the MEK and PI3K pathways, with both pathways providing a bypass escape mechanism for the other when either is individually targeted [66]. We previously identified a synergistic antiproliferative response to the combined treatment of NSCLC cell lines with PI3K–mTOR and MEK inhibitors [67]. Additionally, studies have demonstrated that KRAS G12C inhibitors suppress MAPK signalling for a short duration, with the reactivation of the signalling pathway often observed within 24 to 72 h of treatment in cell lines and mice xenograft models [68]. This signalling rebound is a result of the compensatory activation of receptor tyrosine kinases (RTKs), which is cell-type-specific. Epithelial cells typically activated ERBB signalling, while mesenchymal cells activated FGFR or AXL signalling [68]. Co-targeted approaches inhibiting RTKs and mutant KRAS may be more effective than single-agent KRAS inhibitors.

#### 5.1.2. Acquired Resistance

First reports have emerged that have elucidated the mechanisms of clinical resistance through the evaluation of genetic alterations in biopsies and the circulating tumour DNA from the KRYSTAL-1 trial (NCT03785249) [69,70]. Koga et al. identified secondary KRAS mutations causing resistance to sotorasib and adagrasib via the testing of 142 Ba/F3 clones that were resistant to either inhibitor [71]. Other examples of KRAS resistance are through the activation of bypass signalling pathways enabling the continued signalling of the MAPK and/or PI3K pathways, even with the total inhibition of KRAS [72].

A study by Tanaka et al. identified ten distinct resistance alterations that emerged in a patient treated with adagrasib [69]. All 10 alterations converged on the activation of RAS–MAPK signalling, suggesting a central common mechanism of acquired resistance [69]. RAS reactivation occurred via the activation of the NRAS isoform, KRAS-activating mutations in trans (G13D and G12V), the potential loss of KRAS G12C through a mutational switch to a different KRAS mutation in cis, and a novel secondary alteration in KRAS (Y96D) that alters drug binding (Figure 4). KRAS Y96D also confers resistance to other KRAS G12C-selective inhibitors in clinical development [69]. Similar to EGFR inhibition, a histologic transformation from lung adenocarcinoma to squamous cell carcinoma was also observed. Interestingly, these data suggest that adagrasib treatment results in the evolution of a diverse set of adaptive mechanisms in KRAS G12C-mutated cancers, instead of the dominant adaptive mechanisms seen with other targeted therapies. Thus, monitoring for the emergence of drug resistance is more complicated.

Polyclonal-acquired resistance mechanisms to adagrasib treatment have been identified in one patient [68]. These include resistance alterations in four genes (KRAS, NRAS, BRAF, and MAP2K1), all of which converge to reactivate RAS–MAPK signalling. In addition, a novel KRAS Y96D mutation affecting the switch II pocket, to which adagrasib and other inactive-state inhibitors bind, was identified that interferes with protein–drug interaction [53]. 

## 6. Overcoming Drug Resistance and the Future of KRAS Targeting

### 6.1. Synergistic Drug Combinations to Overcome Resistance to KRAS Targeted Therapy

Therapeutic combination strategies to overcome drug resistance include targeting upstream regulators including SHP2, SOS, and RTKs [72,73]. The SHP2 domain is downstream of an RTK; therefore, its inhibition may block the phosphorylation and reactivation of KRAS. SHP2 inhibitors can suppress KRAS G12C GTPase activity and increase the effect of G12C inhibitors, as they increase the KRAS–GDP occupancy and are currently being trialled in combination with sotorasib (NCT04185883) [74] and other KRAS G12C inhibitors.

SOS is responsible for the activation of KRAS through the GDP–GTP exchange; hence, its inhibition would stop its binding to KRAS and terminate any signalling. The SOS inhibitor BI1701963 is currently being tested in combination with the MEK inhibitor trametinib (NCT04111458) [65].

Targeting RTKs in combination with a KRAS G12C inhibitor is a promising direct approach to circumvent the reactivation of KRAS and its downstream effectors induced by RTKs in response to the inhibition of KRAS [71]. The inhibition of downstream targets including MEK and ERK is an effective combination strategy to enable a more sustained response to treatment.

Combinations with cell-cycle checkpoint inhibitors, antiapoptotic inhibitors, immunotherapy, metabolic inhibitors, and co-targeting synthetic lethal genes are also promising treatment strategies. Cyclin-dependent kinases (CDKs) 4/6 have key roles in regulating the cell cycle and can interact with KRAS through the MAPK and PI3K pathways. The inhibitors of CDK4/6 such as palbociclib can amplify the effects of G12C inhibitors [75]. A recent study showed synergy between DT2216 (a clinical-stage BCL-XL PROTAC) and sotorasib in KRAS G12C-mutated NSCLC, CRC, and pancreatic cancer preclinical models [76].

KRAS-mediated signalling plays an important role in the formation of an immunosuppressive tumour microenvironment (TME) and the modification of immune cells. In a preclinical study, sotorasib treatment significantly induced the infiltration of CD8+ T cells, macrophages, and dendritic cells in the TME [15]. Sotorasib contributed to the formation of a proinflammatory microenvironment and enhanced immunosurveillance. Similarly, adagrasib decreased intertumoral immunosuppressive myeloid-derived suppressor cells and M2-polarised macrophages and increased immune-promoting M1-polarised macrophages, dendritic cells, CD4+ cells, and NKT cells in KRAS G12C tumours [77]. The effects of combining KRAS G12C inhibitors with immunotherapy are currently being tested in clinical trials (NCT03600883) with sotorasib [70] and with adagrasib (NCT03785249) [72].

A recent study has shown that the paralogues HRAS and NRAS are specific suppressors of oncogenic KRAS-driven lung cancer [78]. The expression of these paralogues differs between cell types and cancer types; thus, it is likely that the combined stoichiometry of KRAS, HRAS, and NRAS proteins drives the cellular and in vivo phenotypes. Further investigations are required to determine whether modulating RAS protein interactions or forcing interparalogue competition may lead to novel therapeutic intervention strategies [78].

### 6.2. Co-Targeting Metabolic Pathways in KRAS Mutant Lung Cancer

The ability of KRAS mutant tumours to regulate cancer-cell metabolism is well documented and provides a rationale for co-targeting strategies [79]. In this regard, a recent publication by us described the altered expression of ACOX2 in NSCLC [80]. This protein is more commonly associated with peroxisomal-based processes, including the breakdown and degradation of branched-chain fatty acids and bile acid biosynthesis [81,82], but peroxisomes themselves play many diverse roles in biological processes ranging from cellular lipid metabolism to the synthesis of cellular signalling molecules and response to cellular stress [83,84]. It is interesting to note that the links between peroxisomal-based processes, the TCA cycle, and metabolic rewiring were first described for KRAS-mutated lung cancer [85], and additional links between this oncogenic driver mutation and the key regulators of metabolic processes have now been described [79].

In addition to NSCLC, ACOX2 has also been shown to be dysregulated in various cancers, including breast cancer [86,87], prostate cancer [88,89], and hepatocellular carcinoma [90], and targeting peroxisomes is an emerging potential therapeutic approach including NSCLC [91,92,93].

A role for KRAS mutations and peroxisomes has recently been identified in peroxisome proliferator-activated receptors (PPARs), which were originally identified as the receptors capable of inducing the proliferation of peroxisomes in cells [94]. In this regard, the mutated KRAS G12D was shown to induce a rapid acceleration of KRAS-mutated pancreatic carcinogenesis via PPARγ [95], and also to promote intestinal adenoma formation in the large intestine of APC mice via PPARδ [96]. However, analysis using TIMER 2.0 [97] shows that the levels of PPARγ mRNA are not altered in KRAS-mutated NSCLC compared with wild-type, while the expression of PPARδ appears to be decreased in the LUAD subtype when KRAS is mutated.

An analysis of the effect of KRAS mutation on the expression of ACOX2 in TCGA datasets for commonly mutated cancers (Figure 5A) clearly demonstrates significant upregulation of ACOX2 mRNA in both LUAD (Figure 5B) and LUSC (Figure 5C) subtypes. Moreover, analysis in cancers that are frequently mutated for KRAS also demonstrates an upregulated expression of ACOX2 mRNA in KRAS-mutated colon adenocarcinoma (Figure 5E) or rectal adenocarcinoma (Figure 5F), with the exception of pancreatic adenocarcinoma, where ACOX2 expression was decreased as a consequence of KRAS mutation (Figure 5D). These results suggest that ACOX2 could potentially be a target for therapy in KRAS-mutated cancers. In our analysis of ACOX2 in NSCLC [80], we identified that ACOX2 may also be a target for MEK inhibitors, which are currently being investigated for their potential utility in the treatment of KRAS-mutated NSCLC [98,99]. Other targets identified by us included potential sensitivity to HSP90 inhibitors and crizotinib [80], whilst others have linked ERK inhibitors to enhanced sensitivity in KRAS-mutated cancer [98]. These results suggest that the peroxisomal pathway could be an area worthy of future investigation in particular for KRAS-mutated cancer.

In addition to peroxisomal-targeting strategies, recent studies have also linked glutamine metabolism to KRAS-mutated cancers. For instance, KRAS-mutated pancreatic cancer results in the upregulation and activation of NRF2 (NFE2L), a master regulator of the antioxidant network resulting in the metabolically rewiring and elevation of the pathways involved in glutamine metabolism [100] and a key pathway in NSCLC alongside its negative regulator KEAP1 [30,101]. Critically, the elevated NRF2 resulted in enhanced chemoresistance to gemcitabine, which could be abrogated by treating cells with the drugs that affect glutaminolysis [95]. In this regard, the co-mutations of KEAP1/NRF2 (NFE2L) were found in 27% of KRAS-mutated NSCLC [23]. Critically, KRAS-mutated NSCLC has now been shown to be glutamine-dependent [102] and sensitive to the glutaminase inhibitor CB839 (telaglenastat). A Phase I trial of telaglenastat in combination with the mTOR inhibitor sapanisertib in patients with advanced NSCLC is currently underway (Clinicaltrials.Gov: NCT04250545) [103], the results of which will be eagerly awaited.

## 7. Conclusions

The targeting mutant KRAS is an attractive therapeutic strategy due to its high prevalence across tumour types and its role in initiating and sustaining tumour growth. Once thought of as ‘undruggable’ due to the lack of classic drug-binding sites, the recent approval of KRAS G12C inhibitors sotorasib and adagrasib in locally advanced or metastatic NSCLC has set a new paradigm for the treatment of KRAS G12C-mutated cancers.

The development of directly targeted inhibitors to other KRAS mutants is still ongoing with the KRAS G12D allele-specific inhibitor (MRTX-1133) receiving IND clearance by the US FDA (January 2023) enabling Phase I initiation. Although the initial single-agent response data are promising, overcoming the multiple parallel drug resistance mechanisms that emerge quickly in response to KRAS inhibition needs to be managed. Combination-targeted approaches will be required to provide patients with a more durable response to treatment. Likewise, novel delivery systems such as NPs and liposomes may help to overcome the difficulties of infiltrating drug-resistant tumours such as IMAs. Finally, the emergence of new strategies such as cancer vaccines, adoptive T-cell therapy, and pan-KRAS PROTACs will all likely provide further treatment options to overcome KRAS-mutated cancer in the future. 

## Figures and Tables

**Figure 1 cancers-15-01635-f001:**
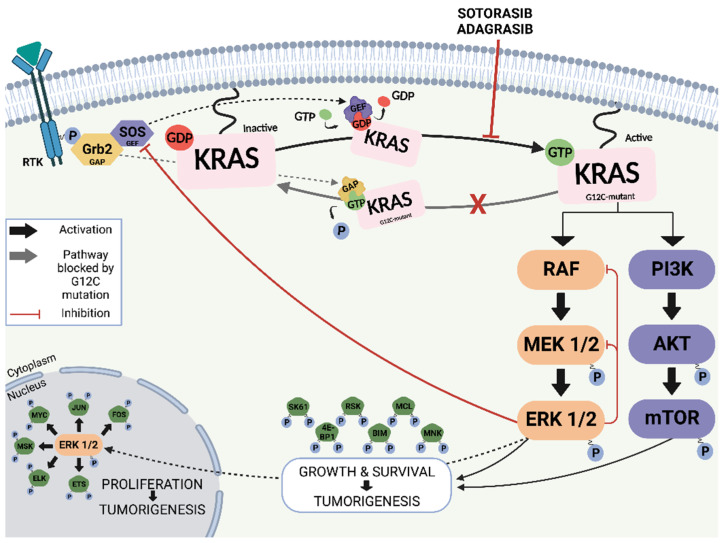
Oncogenic signalling pathways of G12C-mutated KRAS and inhibition by direct inhibitors. Upon receptor tyrosine kinase (RTK) activation, GEFs bind to KRAS and facilitate the exchange of bound-GDP for GTP, thus switching KRAS to its active state. Active KRAS induces signal transduction through MAPK and PI3K pathways, promoting cell proliferation, growth and survival. The G12C mutation blocks GAP binding to KRAS, thus inhibiting GTP hydrolysis and locking G12C-mutant KRAS in its active state. This leads to constitutive activation of the MAPK and PI3K signaling pathways, thus promoting tumorigenesis. Direct KRAS G12C inhibitors (sotorasib and adagrasib) bind GDP bound KRAS and keep it inactive. Image generated using Biorender.com (accessed on 12 January 2023).

**Figure 2 cancers-15-01635-f002:**
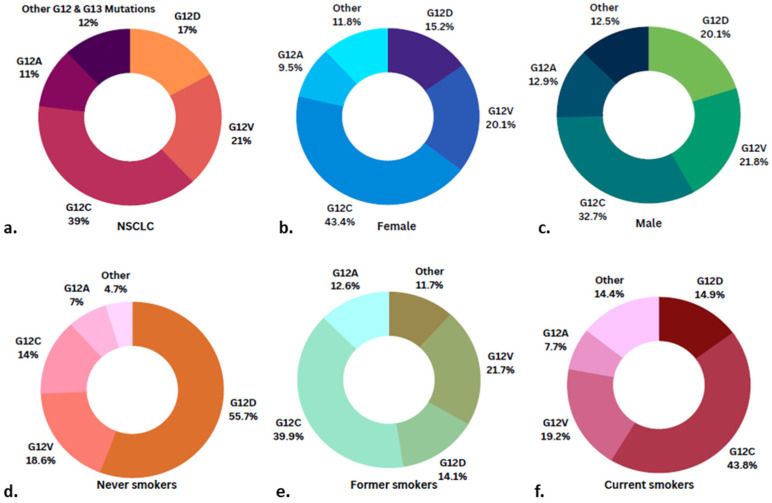
Types and prevalence of KRAS mutations in NSCLC adenocarcinoma. In a study of 3026 patients (1128 males (37%) and 1898 females (63%)) with NSCLC adenocarcinoma, 670 patients harboured KRAS mutations. Pie charts demonstrate the frequency of different KRAS mutation subtypes in (**a**) the entire cohort, (**b**) females (*n* = 422), (**c**) males (*n* = 248), (**d**) never smokers (*n* = 43), (**e**) former smokers (*n* = 419), and (**f**) current smokers (*n* = 208) [13].

**Figure 3 cancers-15-01635-f003:**
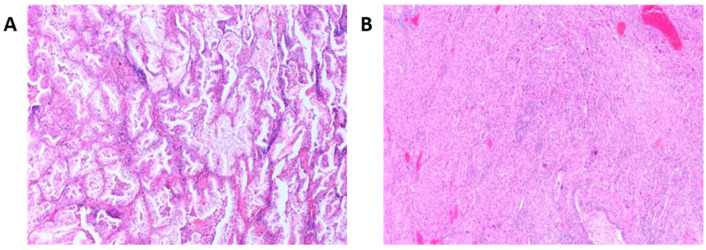
H&E images of (**A**) invasive mucinous adenocarcinoma and (**B**) non-mucinous adenocarcinoma of the lung. Invasive mucinous carcinomas have goblet morphology with abundant intracytoplasmic mucin.

**Figure 4 cancers-15-01635-f004:**
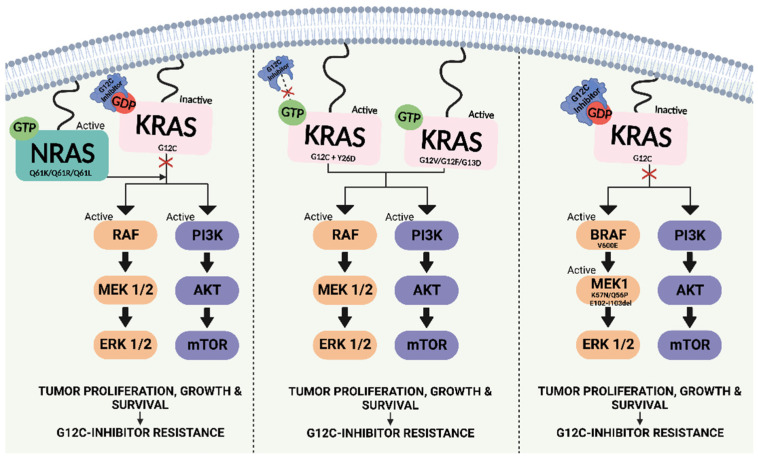
Acquired resistance mechanisms to KRAS G12C inhibitors. Image generated using Biorender.com (accessed on 12 January 2023).

**Figure 5 cancers-15-01635-f005:**
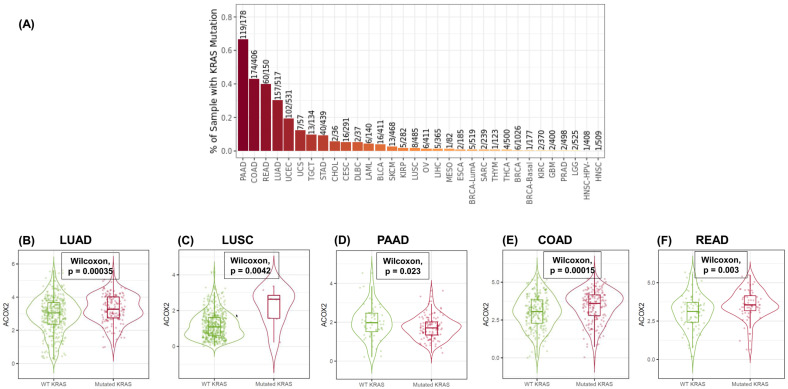
Linking ACOX2 expression to mutated KRAS: (**A**) figure shows prevalence of KRAS mutation across various TCGA datasets. Using Timer 2.0 [97], we analysed the expression of ACOX2 against KRAS (WT vs. mutated) and demonstrate significantly altered expression in (**B**) LUAD (elevated), (**C**) LUSC (elevated), (**D**) PAAD (decreased), (**E**) COAD (elevated), and (**F**) READ (elevated). The text in the image generated byTimer 2.0 was adjusted for clarity.

**Table 1 cancers-15-01635-t001:** Overview of KRAS G12C-directed inhibitors in clinical trials.

Compound(s)	Company	Cancer Type Tested	Combinations	NCT Number
AMG 510/sotorasib * CodeBreak 100 Ph1/2 CodeBreak 101 Ph1b CodeBreak 200 Ph3 A lung- MAP treatment Ph2	Amgen	NSCLC CRC Solid tumours incl. PAC Only KRAS G12C mutation	Monotherapy PD-1/PD-L1 EGFR TKI Chemotherapy EGFR Ab ± chemo or + MEKi VEGF Ab ± Chemo SHP-2 mTOR CDK inhibitor	NCT03600883 NCT04185883 NCT04303780 NCT04625647
MRTX849/adagrasib * KRYSTAL-1 Ph1/2 KRYSTAL-2 Ph1/2 KRYSTAL-7 Ph2 KRYSTAL-10 Ph3 KRYSTAL-12 Ph3	Mirati	NSCLC CRC Solid tumours incl. PaC Only KRAS G12C mutation	Monotherapy PD-1 EGFR Ab EGFR TKI SHP-2 CDK inhibitor SOS1	NCT03785249 NCT04330664 NCT04613596 NCT04793958 NCT04685135
ARS-3248/ JNJ-74699157 Ph1	Wellspring Biosciences J&J	Solid tumours KRAS G12C mut	Monotherapy	NCT04006301 Terminated after 10 pts enrolled
LY3499446 Ph1/2	Eli Lily	NSCLC CRC Solid tumours KRAS G12C mutation	Monotherapy CDK inhibitor EGFR Ab EGFR TKI	NCT04165031 Terminated due to toxicity
GDC-6036 Ph1	Genentech/Roche	NSCLC, CRC solid tumours incl. PaC Only KRAS G12C mutation	Monotherapy PD-L1 EGFR Ab VEGF Ab EGFR TKI	NCT04449874
D-1553 Ph1/2 D-1553 Ph1/2	InventisBio	Solid tumours KRAS G12C mut	Monotherapy	NCT04585035 NCT05383898
JDQ443 Ph1/2	Novartis	Solid tumours KRAS G12 mut	SHP-2 PD-1	NCT04699188
BI 1,823,911 Ph1a/1b	Boehringer Ingelheim	Solid tumours KRAS G12 mut	Monotherapy + BI 1701963	NCT04973163
LY3537982 Ph1a/1b	Eli Lilly	Solid tumours KRAS G12 mut	Monotherapy CDK inhibitor EGFR TKI PD1 ERK1/2 inhibitor Aurora A EGFR Ab SHP-2	NCT04956640
JAB-21822 Ph1/2 JAB-21822 Ph 1b/2 JAB-21822 Ph 1/2 JAB-21822 Ph 1/2a	Jacobio	Solid tumours KRAS G12 mut	Monotherapy EGFR Ab SHP-2	NCT05009329 NCT05194995 NCT05002270 NCT05288205
YL-15293 Ph 1/2	Shanghai YinLi	Solid tumours KRAS G12 mut	Monotherapy	NCT05119933
RMC-6291 Ph1/1b	Revolution Medicine	Solid tumours KRAS G12 mut	Monotherapy	NCT05462717

* Approved for use in the clinic.

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
