# Peer review of "Treatment Strategies for KRAS-Mutated Non-Small-Cell Lung Cancer"

_cancers, 2023, doi:10.3390/cancers15061635_

Round 1

Reviewer 1 Report (Previous Reviewer 1)

All concerns have been addressed, ready for acceptance. 

Author Response

Thank you for taking the time to review and your positive feedback.

Reviewer 2 Report (Previous Reviewer 2)

Dear Editor and Authors,

Thank you for asking me to re-evaluate this previously submitted review manuscript. I was initially skeptical of its value to the scientific community and had voiced this concern in my previous evaluation. Reading it initially was very dry, unimaginable and basically read like an old textbook – old and moldy!!

I am happy to see/say that the authors have improved their work significantly and it is now much better.

Thank you,   

Author Response

Thank you for taking the time to review and your positive feedback.

Reviewer 3 Report (Previous Reviewer 3)

Comments: The manuscript is so interesting. However, very few points should be clarified before being accepted. 

1- Authors should have to discuss if there is a relation between KRAS and other RAS GTPases such as HARS and NRAS.

2- The manuscript lacks information if the mutated KRAS is associated only with non-small lung cancer or can be expressed in other lung cancer types.

3- In Fig.3 (A and B) were not written in the figure. 

4- Few English typos errors should be revised thoroughly over all the text. 

Author Response

Dear Reviewer

Thank you for take the time to review our manuscript and your valuable feedback. We have now addressed you comments and trust all is in order.

1- Authors should have to discuss if there is a relation between KRAS and other RAS GTPases such as HARS and NRAS.

We have know highlighted that KRAS, HRAS and NRAS are all paralogues and that HRAS and NRAS are suppressors of KRAS-driven lung tumour growth. 

2- The manuscript lacks information if the mutated KRAS is associated only with non-small lung cancer or can be expressed in other lung cancer types.

We have information on KRAS mutation rates in SCLC.

3- In Fig.3 (A and B) were not written in the figure. 

We have added A & B to figure.

4- Few English typos errors should be revised thoroughly over all the text. 

We have checked for typos and corrected.

Round 2

Reviewer 3 Report (Previous Reviewer 3)

Manuscript was revised point by point according to reviewer comments and now it is  more acceptable 

This manuscript is a resubmission of an earlier submission. The following is a list of the peer review reports and author responses from that submission.

Round 1

Reviewer 1 Report

A timely review article by Dr. Gately and group elaborating on the various Treatment strategies for KRAS mutant lung cancers. This a very well-written review article with updated information that stands out from the other published review in the same field. Though few things must be addressed, as mentioned below. 

1. It has been discussed how KRAS regulates cancer metabolism and which also opens up novel therapeutic strategies (PMID: 33870211 and PMID: 28570035). The authors should add a few lines on this aspect.

2. It has been shown recently that inhibition of glutamine metabolism activated may be an effective treatment strategy for KRAS-driven cancers (PMID: 31911550 and PMID: 33137541). This topic must be discussed by adding a few lines.

3. Fig 3 figure labels are not clearly readable. Please adjust those. 

4. Authors might add a pie chart showing the percentage of KRAS mutation in various types of lung cancer, including NSCLC.

Reviewer 2 Report

Dear Editor and Authors,

Thank you for asking me to evaluate this review manuscript on the treatment strategies for KRAS mutant Non - Small Cell Lung Cancer. This is an extensive work, quite thorough and well structured. The language is clear and understandable with only minor spelling and expression mistakes.

However, my main point of concern (and game ender!!) is that it is quite long and verbose/loquacious providing much information which at points are repetitive and which, quite importantly, are well known to the oncological community and previously presented. Therefore, it does not offer new information nor does it have a specific focus on which the reader can grab on! It felt like reading a textbook as opposed to a review and unfortunately a tedious one at that!!

In truth I cannot see the benefit to the oncological community for this work to be put out there as the knowledge is known and available!! Therefore, I am not supportive of publication I am sorry to say!! I wish all well however.

Reviewer 3 Report

Comments: The work is an interesting manuscript, well written and in the scope of Cancer journal. However there are few issues  should be clarified

1.     Scheme for signalling pathway to illustrate how monotherapy can target KRAS should have been investigated

2.     Authors didn't discuss if any of nanoparticles loaded by KRAS inhibitors were reached clinical trials.

3. The mechanism by which nanoparticles can be moved specifically to be accumulated  in lung cancer was not described 

4.     Refs should to be revised according to format of cancer MDPI journal